# LEARNING TO ACT WITHOUT ACTIONS

**Dominik Schmidt**[*]
Weco AI

**Minqi Jiang**
FAIR at Meta AI

## ABSTRACT

Pre-training large models on vast amounts of web data has proven to be an effective approach for obtaining powerful, general models in domains such as language and vision. However, this paradigm has not yet taken hold in reinforcement learning. This is because videos, the most abundant form of embodied behavioral data on the web, lack the action labels required by existing methods for imitating behavior from demonstrations. We introduce **Latent Action Policies** (LAPO), a method for recovering latent action information—and thereby latent-action policies, world models, and inverse dynamics models—purely from videos. LAPO is the first method able to recover the structure of the true action space just from observed dynamics, even in challenging procedurally-generated environments. LAPO enables training latent-action policies that can be rapidly fine-tuned into expert-level policies, either offline using a small action-labeled dataset, or online with rewards. LAPO takes a first step towards pre-training powerful, generalist policies and world models on the vast amounts of videos readily available on the web. Our code is available here: https://github.com/schmidtdominik/LAPO.

## 1 INTRODUCTION

Training on web-scale data has shown to be an effective approach for obtaining powerful models with broad capabilities in domains including language and vision (Radford et al., 2019; Caron et al., 2021). Much recent work has thus sought to apply the same paradigm in deep reinforcement learning (RL, Sutton & Barto, 2018), in order to learn generalist policies from massive amounts of web data (Baker et al., 2022; Reed et al., 2022). However, common methods for learning policies from offline demonstrations, such as *imitation learning* (Pomerleau, 1988; Ross & Bagnell, 2010) and *offline RL* (Kumar et al., 2020; Levine et al., 2020), typically require action or reward labels, which are missing from purely observational data, such as videos found on the web.

In this work we introduce **Latent Action Policies** (LAPO), a method for recovering latent-action information from videos. In addition to inferring the structure of the underlying action space purely from observed dynamics, LAPO produces latent-action versions of useful RL models including forward-dynamics models (world models), inverse-dynamics models, and importantly, policies.

LAPO is founded on the key insights that (1) some notion of a *latent action* that explains an environment transition can be inferred from observations alone, and (2) given such inferred latent actions per transition, a *latent-action policy* can be obtained using standard imitation learning methods. Our experiments provide strong evidence that such latent policies accurately capture the observed expert's behavior, by showing that they can be efficiently fine-tuned into expert-level policies in the true action space. This results makes LAPO a significant step toward pre-training general, rapidly-adaptable policies on massive action-free demonstration datasets, such as the vast quantities of videos available on the web.

Crucially, LAPO learns to infer latent actions, and consequently, obtain latent action policies in a fully unsupervised manner. LAPO is similar to prior work (Torabi et al., 2018; Schmeckpeper et al., 2020; Baker et al., 2022; Zheng et al., 2023) in that we first train an *inverse dynamics model* (IDM) that predicts the action taken between two consecutive observations, and then use this IDM to add action labels to a large action-free dataset. However, unlike these prior works, which rely on some amount of ground-truth action-labelled data for training the IDM, LAPO does not make use of any labels and infers latent action information purely from observed environment dynamics. To do so,

[*]Correspondence to dominik.schmidt.22@ucl.ac.uk .

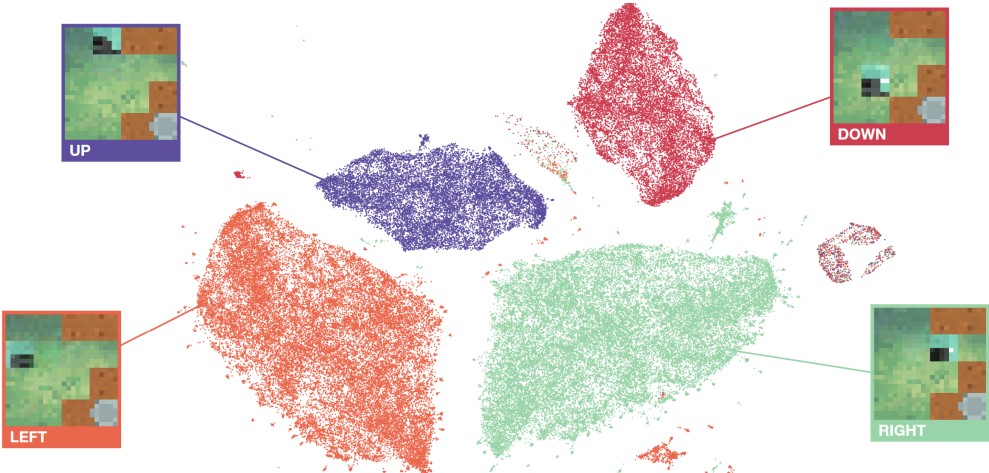

Figure 1: UMAP projection of the learned latent action space for `Miner` alongside illustrative next-state predictions generated by the FDM for each cluster of latent actions. Each point represents a continuous latent action generated by the IDM for a transition in the video dataset. Each point is color-coded by the true action taken by the agent at that transition. For clarity, `NOOP` actions are omitted. The structure of the latent action space is highly interpretable and closely corresponds to the true action space, even though no ground-truth action labels were used during training.

the IDM in LAPO learns to predict *latent actions* rather than true actions. These latent actions take the form of a learned representation that explains an observed transition. To learn this representation, LAPO makes use of a simple unsupervised objective that seeks to establish predictive consistency between the IDM and a separate *forward dynamics model* (FDM). The FDM is trained on transitions consisting of two consecutive observations $(o_t, o_{t+1})$ to predict the future observation $o_{t+1}$, given the past observation $o_t$ and a latent action. This latent action is generated by the IDM given both past and future observations. Thus, unlike the FDM which sees only the past, the IDM has access to both the past and future and learns to pass useful information about the future to the FDM through the latent action. By making the latent action an information bottleneck (Tishby et al., 2000), we prevent the IDM from simply forwarding the entire future observation to the FDM, thereby forcing it to learn a highly compressed encoding of state transitions. We refer to this encoding as a latent action, as it captures the observable effects of the agent's true actions.

In our experiments in Section 6, we train a latent IDM via LAPO on large expert-level action-free offline datasets for each of the 16 games of the Procgen Benchmark (Cobbe et al., 2019; 2020). We observe that the structure of the learned latent action spaces are highly interpretable, often exhibiting clusters that closely align with the true action space (see Figure 1). This is a remarkable result, as our approach uses zero ground-truth action labels and instead recovers all action information purely from observed dynamics. To go beyond merely recovering action information, we then show how we can leverage the learned latent representation to quickly obtain powerful expert-level policies in downstream tasks: We first use the IDM to assign latent action labels to each transition in the same video dataset on which it was trained. We then perform behavior cloning on the resulting action-observation dataset to learn a latent-action policy, which imitates the latent actions predicted by the IDM. Finally, to deploy the policy in the online environment, we seek to turn it from a latent action policy into a policy in the true action space. We show that if a tiny, action-labeled dataset is available, a latent $\rightarrow$ true action decoder can be learned in a supervised manner and used to decode the outputs of the latent policy. This approach is extremely data-efficient, requiring only $\sim$200 labeled transitions to exceed the performance of 4M steps of regular RL via PPO. Alternatively, if an online environment is available, we show that simply fine-tuning the last layers of the latent policy with a general-purpose RL algorithm allows the policy to rapidly adapt to the true action space and recover (or at times exceed) expert performance. These results indicate that latent policies produced by LAPO capture meaningful behavior in the latent action space.

While we take a first step in demonstrating the potential of these latent representations for obtaining powerful policies in Section 6, the **fundamental contribution of this work** is simply showing that comprehensive action information can be recovered from pure video through unsupervised objectives, such as the one proposed herein. This result opens the door to future work that employs representation learning methods like LAPO to train powerful, generalist policies and world models on massive video datasets from the web.

## 2 RELATED WORK

Most closely related to our approach, ILPO (Edwards et al., 2019) aims to infer latent actions via a dynamics modelling objective. However, unlike LAPO, ILPO learns a latent policy rather than an IDM jointly with the FDM, and uses a discrete rather than a continuous latent action space. For each FDM update, ILPO identifies the discrete latent action that minimizes the next-state prediction error, and only optimizes the FDM for that specific action. In practice, this objective can lead to mode collapse, where a feedback loop causes only a small number of discrete latents to ever be selected (Struckmeier & Kyrki, 2023). By using an IDM rather than a discrete latent policy, LAPO avoids enumerating all hypothetically possible transitions and instead learns the latent action corresponding to the actual observed transition. To train the policy, ILPO minimizes the difference between the true next state and the expectation of the next state under the policy. This loss is ill-conditioned, as even when it is minimized, the next-state predictions for individual latent actions do not have to align with the environment. Moreover, LAPO's use of continuous rather than finite discrete latents may allow better modeling in complex and partially-observable environments, as any information useful for prediction can be captured within the latent representation. We provide evidence for this in Appendix A.5, where we show that ILPO breaks down when applied to more than a single Procgen level at a time. Finally, ILPO's use of discrete latents makes its computational complexity linear rather than constant in the latent dimensionality, due to the required enumeration of all actions.

In the model-based setting, FICC (Ye et al., 2023) pre-trains a world model from observation-only demonstrations. FICC thus seeks to distill dynamics from demonstrations, rather than behaviors, as done by LAPO. Moreover, while the cyclic consistency loss used by FICC closely resembles the LAPO objective, the adaptation phase of FICC must operate in reverse, mapping true actions to latent actions in order to finetune the world model. Unlike the approach taken by LAPO, this adaptation strategy cannot make direct use of online, ground-truth action labels, and requires continued training and application of a world model. In contrast, LAPO directly produces latent action policies imitating the expert behavior, which can be rapidly adapted online to expert-level performance. Outside of RL, Playable Video Generation (PVG, Menapace et al., 2021) similarly focuses on world model learning with a similar objective for the purpose of controllable video generation. Like ILPO, PVG uses a more limiting, small set of discrete latent actions.

A related set of semi-supervised approaches first train an IDM using a smaller dataset containing ground-truth action labels, and then use this IDM to label a larger action-free dataset, which can subsequently be used for behavior cloning. VPT (Baker et al., 2022) and ACO (Zhang et al., 2022) follow this approach, training an IDM on action-labeled data which is then used to generate pseudo labels for training policies on unlabeled footage gathered from the web. RLV (Schmeckpeper et al., 2020) uses an observation-only dataset within an online RL loop, where action and reward labels are provided by an IDM trained on action-labelled data and a hand-crafted reward function respectively. Similarly, BCO (Torabi et al., 2018) trains an IDM through environment interaction, then uses the IDM to label action-free demonstrations for behavior cloning. However, relying on interactions for labels can be inefficient, as finding useful labels online may itself be a difficult RL exploration problem. SS-ORL (Zheng et al., 2023) is similar to VPT but performs offline RL instead of imitation learning on the IDM-labelled data and requires reward-labels. Unlike these methods, LAPO avoids the need for significant amounts of action-labeled data to train an IDM, by directly inferring latent actions and latent policies which can easily be decoded into true actions.

LAPO differs from previous methods for *imitation learning from observation* (IfO; Torabi et al., 2019a;b; Yang et al., 2019), which typically require the imitating policy to have access to the true action space when training on action-free demonstrations. Crucially, unlike these prior approaches, LAPO does not require access to the ground-truth action space to learn from action-free demonstrations. Other methods leverage observation-only demonstrations for purposes other than learning useful behaviors. Intention-Conditioned Value Functions (ICVF; Ghosh et al., 2023) uses a temporal difference learning objective based on observation-only demonstrations to learn state representations useful for downstream RL tasks. Aytar et al. (2018) propose to use expert demonstrations during online RL to guide the agent along the expert's trajectories. For this, they propose two auxiliary classification tasks for learning state representations based on observation-only demonstrations.

## 3 BACKGROUND

### 3.1 REINFORCEMENT LEARNING

Throughout this paper, we consider RL under partial observability, using the framework of the partially-observable Markov decision process (POMDP, Åström, 1965; Kaelbling et al., 1998). A POMDP consists of a tuple $(\mathcal{S}, \mathcal{A}, \mathcal{T}, \mathcal{R}, \mathcal{O}, \Omega, \gamma)$, where $\mathcal{S}$ is the state space, $\mathcal{A}$ is the action space, $\mathcal{O}$ is the observation space, and $\gamma$ is the discount factor. At each timestep $t$, the RL agent receives an observation $o_t$ derived from the state $s_t$, according to the observation function $\Omega : \mathcal{S} \mapsto \mathcal{O}$, and takes an action according to its policy $\pi(a_t|o_t)$, in order to maximize its expected discounted return, $\sum_{k=t}^{\infty} \gamma^{k-t} r_k$. The environment then transitions to the next state $s_{t+1}$ according to the transition function, $\mathcal{T} : \mathcal{S} \times \mathcal{A} \mapsto \mathcal{S}$, and agent receives a reward $r_t$ based on the reward function $\mathcal{R} : \mathcal{S} \times \mathcal{A} \times \mathcal{S} \mapsto \mathbb{R}$. This work considers first learning a policy $\pi$ from offline demonstrations, followed by further fine-tuning the policy online as the agent interacts with its environment.

### 3.2 LEARNING FROM OBSERVATIONS

Often, we have access to recordings of a performant policy, e.g. a human expert, performing a task of interest. When the dataset includes the action labels for transitions, a supervised learning approach called *behavior cloning* (BC, Pomerleau, 1988) can be used to train an RL policy to directly imitate the expert. Consider a dataset $D$ of such expert trajectories within the environment, where each trajectory $\tau$ consists of a list of all transition tuples $(o_0, a_0, o_1), \ldots, (o_{|\tau|-1}, a_{|\tau|-1}, o_{|\tau|})$ in a complete episode within the environment. BC then trains a policy $\pi_{\text{BC}}$ to imitate the expert by minimizing the cross-entropy loss between the policy's action distribution and the expert's action $a^*$ for each observation in $D$: $\mathcal{L}_{\text{BC}} = -\frac{1}{|D|} \sum_{\tau \in D} \sum_{t=0}^{|\tau|} \log(\pi(a_t^*|o_t))$.

Unfortunately, most demonstration data in the wild, e.g. videos, do not contain action labels. In this case, the demonstration data simply consists of a continuous stream of observations taking the form $(o_0, o_1, \ldots, o_{|\tau|})$. This setting, sometimes called *imitation learning from observations* (IfO, Torabi et al., 2019a), poses a more difficult (and practical) challenge. IfO methods often seek to learn a model that predicts the missing action labels, typically trained on a separate dataset with ground-truth actions. Once labeled this way, the previously action-free dataset can then be used for BC. In this work, we likewise convert the IfO problem into the BC problem. However, instead of relying on access to a dataset with ground-truth action labels, our method directly infers *latent actions* $z_t$ that explain each observed transition $(o_t, o_{t+1})$, with which we train *latent action policies*, $\tilde{\pi}(z_t|o_t)$.

### 3.3 DYNAMICS MODELS

LAPO employs two kinds of dynamics models: The first is the *inverse dynamics model*, $p_{\text{IDM}}(a_t|o_t, o_{t+1})$, which predicts which action $a_t$ was taken by the agent between consecutive observations $o_t$ and $o_{t+1}$. An IDM can be used to label a sequence of observations with the corresponding sequence of actions. The second is the *forward dynamics model*, $p_{\text{FDM}}(o_{t+1}|o_t, a_t)$, which predicts the next observation $o_{t+1}$ given the previous observation $o_t$ and the action $a_t$ taken by the agent after observing $o_t$. The FDM can be used as an approximation of the environment's transition function. In this work, the IDM and FDM are deterministic and implemented as deep neural networks. Unlike a standard IDM, the IDM used by LAPO predicts continuous latent actions, $z_t \in \mathcal{Z}$, where $\mathcal{Z} = \mathbb{R}^n$.

### 3.4 VECTOR-QUANTIZATION

Vector-quantization (Gray, 1984, VQ) is a method for learning discrete features by quantizing an underlying continuous representation. This has been shown particularly useful in deep learning, where VQ enables learning discrete representations while allowing gradients to flow through the quantization step. VQ has been used effectively in many domains including vision (van den Oord et al., 2017; Ramesh et al., 2021b; Huh et al., 2023), audio (Dhariwal et al., 2020; Zeghidour et al., 2022), and model-based RL (Micheli et al., 2023). To quantize a continuous vector $\mathbf{z}$, VQ maintains a codebook $\{\mathbf{c}_1, \mathbf{c}_2, \ldots, \mathbf{c}_m\}$ and maps $\mathbf{z}$ to the closest codebook vector $\mathbf{c}_i$. The straight-through gradient estimator is used to pass gradients through the quantization step (Bengio et al., 2013).

## 4 LATENT ACTION POLICIES

### 4.1 LEARNING A LATENT ACTION REPRESENTATION

We now describe our approach for learning a latent IDM via a forward dynamics modelling objective (see Fig. 2). First, to learn about the action at time $t$, we sample a sequence of observations $o_{t-k}, \ldots, o_t, o_{t+1}$ from our observation-only dataset. Here, $o_t$ and $o_{t+1}$ are the observations before and after the action of interest is taken and observations $o_{t-k}, \ldots, o_{t-1}$ are additional context controlled by hyperparameter $k \geq 0$. The IDM then predicts the latent action $z_t$ based on the full sequence of observations.

$$z_t \sim p_{\text{IDM}}(\cdot | o_{t-k}, \ldots, o_t, o_{t+1})$$

Next, the FDM predicts the post-transition observation $o_{t+1}$ based only on the past observations $o_{t-k}, \ldots, o_t$ and the latent action $z_t$.

$$\hat{o}_{t+1} \sim p_{\text{FDM}}(\cdot | o_{t-k}, \ldots, o_t, z_t)$$

Both models are trained jointly via gradient descent to minimize the next state prediction error $||\hat{o}_{t+1} - o_{t+1}||^2$.

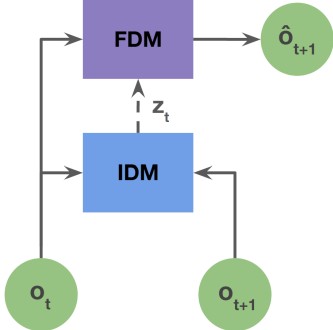

Figure 2: LAPO architecture. Both IDM and FDM observe $o_t$, but only the IDM observes $o_{t+1}$. To enable accurate predictions of $o_{t+1}$, the IDM must pass useful transition information through the quantized information bottleneck $z_t$ to the FDM.

This method, as described so far, would likely result in the IDM learning to simply copy $o_{t+1}$ into $z_t$, and the FDM learning to forward $z_t$ as is. To remedy this, we make the latent action an information bottleneck. This forces the IDM to compress any information to be passed to the FDM. Since both the FDM and IDM have access to past observations but only the IDM observes the post-transition observation $o_{t+1}$, the IDM is learns to encode only the *difference* between $o_{t+1}$ and $o_t$ into $z_t$, rather than full information about $o_{t+1}$. Naturally, one highly efficient encoding of the differences of two consecutive observations, at least in deterministic environments, is simply the agent's true action. Our hypothesis is thus, that forcing the IDM to heavily compress transition information as described, may allow us to learn a latent action representation with a structure closely corresponding to the true action space.

As both the IDM and FDM have access to past observations, the learned latent actions may become conditional on these observations. Intuitively, this means that some latent action $z$ could correspond to different true actions when performed in different states. While this is not necessarily an issue, biasing the method toward simpler representations is likely preferable (Solmonoff, 1964; Schmidhuber, 1997; Hutter, 2003). Consequently, we apply vector quantization (VQ) to each latent action before passing it to the FDM, thus forcing the IDM to reuse the limited number of discrete latents across different parts of the state-space, leading to disentangled representations.

### 4.2 BEHAVIOR CLONING A LATENT ACTION POLICY

Using the trained latent IDM, we now seek to obtain a policy. For this, we initialize a latent policy $\pi : \mathcal{O} \rightarrow \mathcal{Z}$ and perform behavior cloning on the same observation-only dataset that the IDM was trained on, with the required action labels generated by the IDM. This is done via gradient descent with respect to policy parameters on the loss $||\pi(o_t) - z_t||^2$ where $z_t \sim p_{\text{IDM}}(\cdot | o_{t-k}, \ldots, o_t, o_{t+1})$.

### 4.3 DECODING LATENT ACTIONS

The policy $\pi$, obtained via BC, produces actions in the latent action space. We now consider different ways to adapt $\pi$ to produce outputs in the true action space, depending on the kinds of data available.

**Small action-labeled dataset.** If a small dataset of true-action-labeled transitions is available, we can train an action decoder, $d : \mathcal{Z} \rightarrow \mathcal{A}$, to map the IDM-predicted latent action for each transition to the known ground-truth action. The trained decoder can then be composed with the frozen latent policy and the resulting decoded latent policy, $d \circ \pi : \mathcal{O} \rightarrow \mathcal{A}$, can be deployed online.

**Online environment.** By interacting with the online environment we get both action-labeled transitions, which can be used to train a decoder in a supervised manner as described above, and a reward signal, which can be used for online RL. By doing both simultaneously, we can quickly bootstrap a crude, initial decoder via supervised learning, and then fine-tune the entire composed policy $d \circ \pi$ by directly optimizing for the environment's reward function via RL. Through RL, the performance of the policy can be potentially be improved beyond that of the data-generating policy. In particular, when the data was generated by a mixture of policies of different skill levels or pursuing different goals (as is the case with data from the web), this step can extract the data-generating policies more closely aligned with the reward function used for fine-tuning.

## 5 EXPERIMENTAL SETTING

Our experiments center on the Procgen Benchmark (Cobbe et al., 2020), as it features a wide variety of tasks that present different challenges for our method. Compared to other classic benchmarks such as Atari (Bellemare et al., 2013), the procedural generation underlying Procgen introduces a difficult generalization problem and results in greater visual complexity which makes dynamics modeling at pixel-level challenging. Moreover, several Procgen environments feature partial observability, which along with stochasticity, presents issues for methods attempting to infer actions purely from observation by making it ambiguous which parts of an environment transition are due to the agent and which are due to stochastic effects or unobserved information. Our observation-only dataset consists of approximately 8M frames sampled from an expert policy that was trained with PPO for 50M frames. The use of this synthetic dataset, rather than video data from the web, allows us to better evaluate our method, as we can directly access the expert policy's true actions, as well as metadata such as episodic returns for evaluation purposes.

We use the IMPALA-CNN (Espeholt et al., 2018) to implement both our policy and IDM with a 4x channel multiplier as used by Cobbe et al. (2020), and U-Net (Ronneberger et al., 2015) based on a ResNet backbone (He et al., 2016) with approximately 8M parameters for the FDM. The latent action decoder is a fully-connected network with hidden sizes $(128, 128)$. We use an EMA-based update (Polyak & Juditsky, 1992) for the vector quantization embedding. We use a single observation of additional pre-transition context, i.e. $k = 1$. When decoding the latent policy in the online environment, we have two decoder heads on top of the latent policy whose logits are averaged per action before applying softmax. One head is trained in a supervised manner on $(a_t, z_t)$ tuples from historical data, the other is trained via Proximal Policy Optimization (PPO, Schulman et al., 2017). We keep all convolutional layers frozen and found that a much larger learning rate of 0.01 can be stably used when only training these final few layers. We similarly tuned the learning rate for training the full network from scratch, but found no improvement compared to the original value of 5e-4. Other hyperparameters are given in Appendix A.4.

## 6 RESULTS AND DISCUSSION

### 6.1 DECODING THE LATENT POLICY ONLINE

We now discuss our results for decoding the latent policy in the online environment. Recall, that this is the third stage of our approach, after first training a latent IDM, followed by obtaining a latent policy via behavior cloning. In this experiment, we compare how decoding the latent policy (via supervised learning and RL) compares to learning a policy from scratch purely from environment interaction via PPO. Since our work focuses on the setting where large amounts of observation-only data are freely available, but environment interaction is limited or expensive, we run PPO for 4M steps rather than the 25M steps suggested by Cobbe et al. (2020). Experiments are performed separately per environment. As can be seen in Figure 3, using LAPO's latent policy we are able to fully recover expert performance within only 4M frames, while PPO from scratch reaches only 44% of expert performance in the same period. We highlight that in 9 of 16 tasks, our approach exceeds the performance of the expert (which was trained for 25M frames) within only 4M frames. We also provide results for two ablations. The "RL, no-SL" ablation trains the decoder only using reinforcement learning. In this setting, LAPO can be viewed as an RL-pretraining step that produces an initial policy representation that is useful for downstream RL. The "RL, no-SL, no-VQ" ablation additionally removes vector-quantization, showing that VQ is an important component of our method.

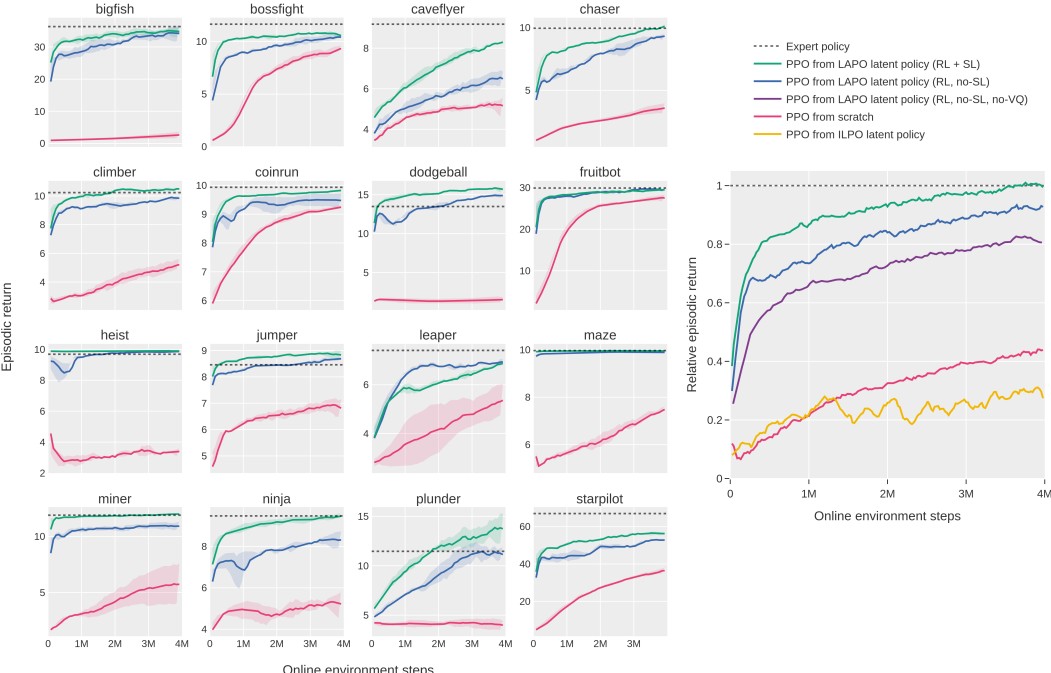

Figure 3: Left: Mean episodic returns (over the course of training) for decoding LAPO's latent policy and PPO from scratch (averaged across 3 seeds). Right: Mean test returns relative to per-environment expert policies averaged across all 16 Procgen environments. Error bars indicate standard deviation across seeds.

We use ILPO as our primary baseline, but found that its policy immediately collapses in several Procgen tasks. When it does not collapse, online decoding of the ILPO policy did not perform better than PPO from scratch. While Edwards et al. (2019) do provide positive results for `CoinRun` (which we reproduce in Appendix A.5), their experiments target only individual levels of this single task. We thus hypothesize that ILPO's policy collapse is in part due to the modeling challenge posed by the visually diverse levels in the full, procedurally-generated setting. By using an IDM rather than a policy, plus a high-dimensional, continuous latent space, LAPO can capture stochasticity and unobserved information (such as off-screen parts of the game). In contrast, it is likely difficult for ILPO to model the full distribution of possible transitions using only a few discrete latents.

## 6.2 DECODING THE LATENT POLICY OFFLINE

Next, we consider training a latent action decoder fully-offline using only a small action-labeled dataset and no access to the online environment. The trained decoder is then composed with the latent policy to map the policy's actions into the true action space. As we will see in Section 6.3, the latent and true action spaces share similar structure. This similarity allows an effective decoder to be trained on only a miniscule amount of action-labeled data. Figure 4 shows that a decoder trained on less than 256 labeled transitions matches the performance of a policy trained from scratch for 4M steps. However, we observe that with increasing dataset size, performance eventually plateaus below the level of online decoder training. This is likely because, as previously noted, latent actions are not necessarily state-invariant. A decoder that only has access to a latent action, but not the state in which that action is being taken, may not always be able to successfully decode it. We provide per-environment results in Appendix A.1.

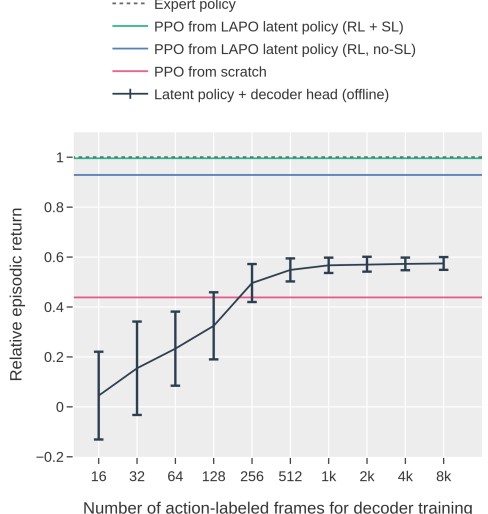

Figure 4: Offline decoding performance vs. # labeled transitions (Mean and std across 3 seeds).

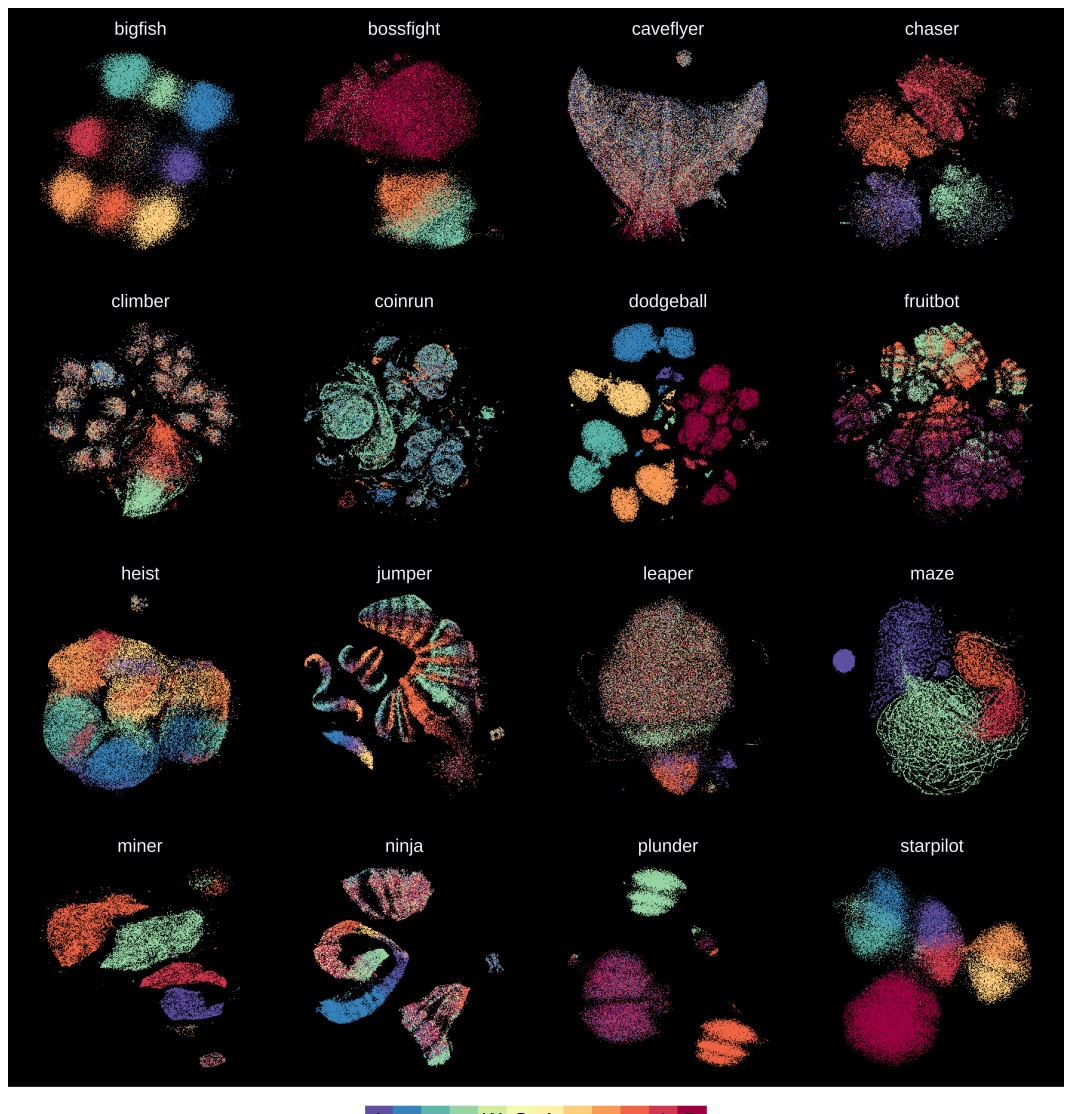

Figure 5: UMAP projection of the learned latent action space for all 16 procgen games. Each point represents the continuous (pre-quantization) latent action generated by the IDM for a transition in the observation-only dataset. Each point is color-coded by the true action taken by the agent at that transition (true action labels are only for visualization, not used for training). Arrows in the legend correspond to movement directions. NOOP actions are omitted for clarity.

## 6.3 INSPECTING THE LATENT ACTION SPACE

To better understand what LAPO learns, we generate a UMAP projection (McInnes et al., 2018) of the latent action space for each of the 16 Procgen environments, shown in Figure 5. For most games, the structure of the latent action space is highly interpretable, with distinct clusters of latent actions closely corresponding to the true discrete actions. We also observe that the structure of the latent space varies greatly across environments. In BigFish, Chaser, Dodgeball, Maze, Miner, Plunder, and StarPilot, latent actions aggregate together in well-defined clusters aligned with the true actions. In other environments, including Climber, CoinRun, FruitBot, Jumper and Ninja, the latent action space exhibits more fragmented structure. This split roughly corresponds to environments with higher or lower degrees of partial observability. In the latter group of environments, the pixel observation shows only part of the level, cropped locally to the player character. Thus, when the player is moving, most observations will show parts of the level that were

hidden in previous observations. In order for the FDM to accurately predict the next observation, the IDM needs to encode this off-screen information into the latent action, which we hypothesize leads to more complex structures in the latent space. However, as demonstrated in Section 6.1, the latent-space policy still performs well downstream even on environments with greater degrees of partial observability. Moreso, we find that vector quantization has a large impact in simplifying the structure of the latent action space. We generate the same UMAP projection for an ablation of our method without vector quantization in Appendix A.2. We note that for several environments, including those with and without partial observability, latent action clusters corresponding to distinct true actions are at times duplicated. For example, in `BigFish`, each true action sees four latent action clusters—likely reflecting how taking the same true action in different parts of the state-space can have differing effects. These results strongly support our hypothesis that vector quantization leads to simpler latent representations with less fragmentation within actions.

### 6.4 LIMITATIONS

A few factors can adversely impact the performance of LAPO. First, actions that have a delayed effect in observations will be predicted to take place with the same delay, i.e. the latent policy actually models the visible effects of an action, not the action itself. Nevertheless, in most environments, actions that have any impact on the state of the environment will elicit some degree of immediate change in the observation. Moreover, delayed actions can be partially addressed by extending the IDM and FDM architecture to consider multiple timesteps into the past and future, e.g. by using a Transformer-based architecture (Vaswani et al., 2017). Second, significant stochasticity can make it difficult for the IDM to compress the useful bits of information among the noise, degrading the quality of the latent representation. This issue can potentially be mitigated by training on much larger datasets. Lastly, training on much larger datasets—as would be required for modeling more complex domains like web-scale video—would require scaling up the model architecture, which introduces new challenges in balancing the strength of the FDM and the capacity of latent actions representations, as is often the case in autoencoding architectures (Chen et al., 2016).

## 7 CONCLUSION

This work introduced LAPO, a method for training policies over a learned latent action space, inferred from purely observational data. Unlike prior work on imitation learning from observation, LAPO does not rely on access to the true action space or a predefined set of discrete latent actions to learn a useful, pretrained policy. Instead, LAPO learns a latent action space end-to-end, by optimizing an unsupervised objective based on predictive consistency between an inverse and a forward dynamics model. Vector quantization of the continuous latent actions induces an information bottleneck that forces the quantized actions to capture state-invariant transition information. Across all 16 games of the challenging Procgen Benchmark, we show that this approach can learn latent action spaces that reflect the structure of the true action spaces, despite LAPO never having access to the latter. We then demonstrate that latent action policies, obtained through behavior cloning of latent actions, can serve as useful pretrained models that can be rapidly adapted to recover or even exceed the performance of the expert that generated the original action-free dataset.

Our results thus suggest that LAPO can serve as a useful approach for obtaining rapidly-adaptable pretrained policies from web-scale action-free demonstration data, e.g. in the form of videos. We believe LAPO could serve as an important step in unlocking the web-scale unsupervised pretraining paradigm that has proven effective in other domains like language and vision (Brown et al., 2020; Radford et al., 2021; Ramesh et al., 2021a). Toward this goal, we are excited about future work aimed at scaling up the world model and policy to more powerful architectures that can consider multiple timesteps and richer observations. Such models may enable efficient learning of generalist latent action policies in complex, multi-task settings, all within a single LAPO instance, while enabling more effective generalization to tasks unseen during training.

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

# A APPENDIX

## A.1 OFFLINE DECODING RESULTS

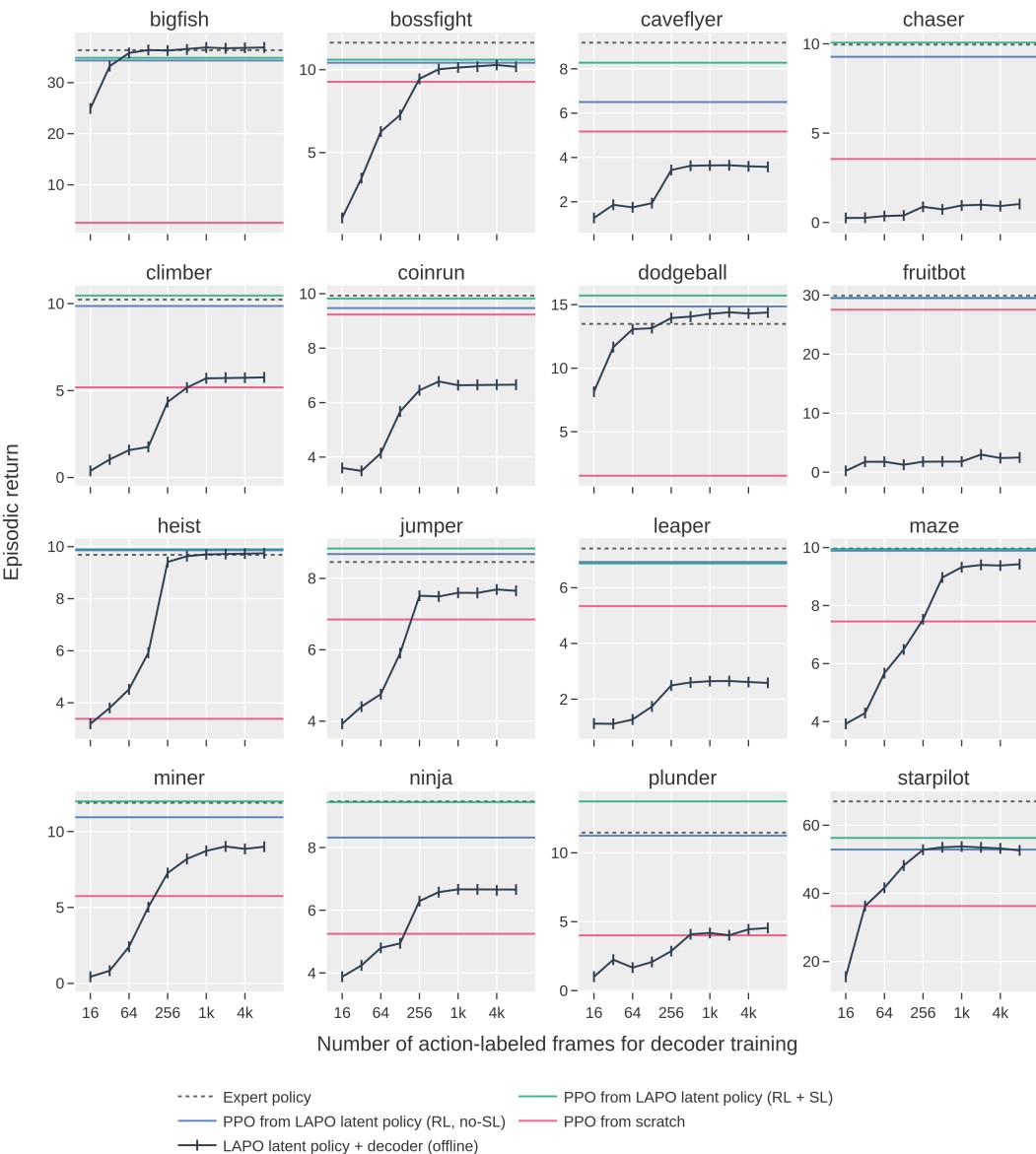

Figure 6: Test performance of the latent policy combined with a latent action decoder that is trained on an action-labelled offline dataset of a certain size, consisting of $(z_t, a_t)$ tuples. An effective decoder can be trained with only a few hundred samples although performance generally plateaus before reaching the performance of our proposed online (RL) decoding approach.

## A.2 LATENT SPACE ANALYSIS FOR NO-VQ ABLATION

Figure 7: UMAP projection of the learned latent action space for all 16 procgen games, generated by IDMs trained without vector-quantization. Each point represents the continuous latent action generated by the IDM for a transition in the observation-only dataset. Each point is color-coded by the true action taken by the agent at that transition (true action labels are only used for visualization, not for training). Arrows in the legend correspond to movement directions. `NOOP` actions are omitted for clarity.

We note that although the No-VQ ablation performs worse in terms of online performance, its FDM loss was significantly lower. This indicates that VQ indeed acts as a bottleneck that constrains the amount of information passed from the IDM to the FDM. By forcing the IDM to compress transition information, this leads to a better latent representation for downstream policy learning.

## A.3 CONTINUOUS CONTROL EXPERIMENTS

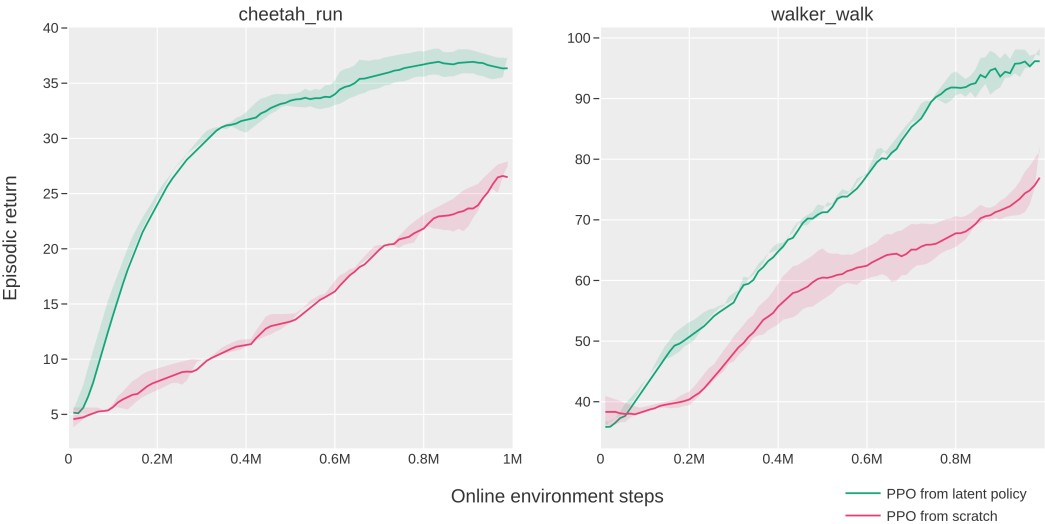

Figure 8: Mean test returns over the course of training for online decoding of a LAPO latent policy compared to PPO from scratch (averaged across 2 seeds) on continuous control tasks. These experiments used the same hyperparameters for training the latent IDM and for behavior cloning as in Procgen experiments.

## A.4 HYPERPARAMETERS

Table 1: Hyperparameters for the three stages of our method. Latent actions are 128-dimensional continuous vectors and are split and quantized into 8 discrete latents with 16-dimensional embeddings. We use Procgen distribution mode `easy`.

| Stage | Parameter | Value |
|---|---|---|
| Latent IDM training | Pre-transition additional context $k$ | 1 |
| | VQ # of codebooks | 2 |
| | VQ # of discrete latents per codebook | 4 |
| | VQ # of embeddings | 64 |
| | VQ embedding dimension | 16 |
| | VQ commitment cost | 0.05 |
| | VQ EMA decay | 0.999 |
| | Learning rate | 2e-4 |
| | Batch size | 128 |
| | Total update steps | 70,000 |
| | IMPALA-CNN channel multiplier | 4 |
| Latent behavior cloning | Learning rate | 2e-4 |
| | Batch size | 128 |
| | Total update steps | 60,000 |
| | IMPALA-CNN channel multiplier | 4 |
| RL online fine-tuning | Total environment interactions | 4,000,000 |
| | PPO hyperparameters | As in (Cobbe et al., 2020). |

## A.5 ILPO REPRODUCTION

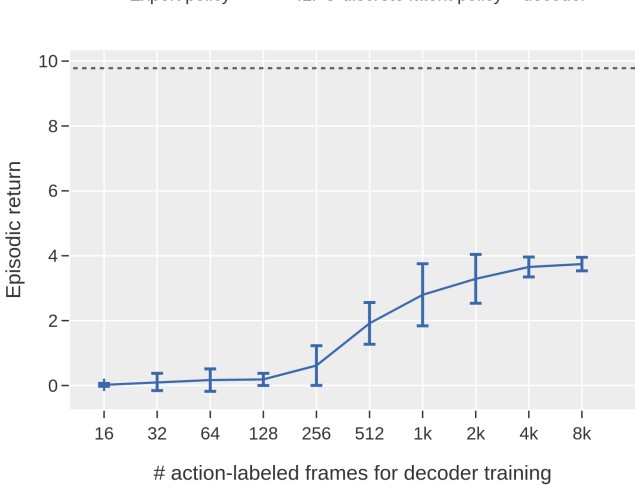

Figure 9: Reproduction of Figure 5.a from Edwards et al. (2019) for a single level from `coinrun`. When applied to data from the full distribution of levels, rather than just a single level, the ILPO FDM consistently collapsed in terms of which discrete latent achieves the minimum of $\mathcal{L}_{\min}$, leading to collapse of the policy too. Results here are not exactly comparable to results from Edwards et al. (2019) since it is unknown to us which specific `coinrun` level was used by the authors.

