# OpenReview forum: "Learning to Act without Actions"
_ICLR.cc/2024/Conference — ICLR 2024 spotlight_

### Official Review · Reviewer_NDab · 2023-10-25

**Soundness:** 3 good
**Presentation:** 3 good
**Contribution:** 2 fair
**Rating:** 6
**Confidence:** 4

**Summary:**

This paper proposes a novel method to pretrain for RL with only observation sequences without action labels. It first infers latent actions, then learns a policy by behavior cloning on the inferred latent actions, and subsequently refining the policy via RL. What sets this research apart from previous studies is its unique strategy in latent action inference, which features a VQ latent action space and a combined optimization target for IDM and FDM.  Experiments on Procgen demonstrate the effectiveness of the proposed method.

**Strengths:**

This paper is well written. Motivation, methodology, and experiments are presented clearly. The idea of optimizing forward modelling objectives using a constrained action latent space is coherent and logical. The provided UMAP projections greatly support the effectiveness of the proposed method.

**Weaknesses:**

1. Toy experimental setup: The paper motivates from learning (pretraining) with large scale web data. However, during experiments, the author uses the data from a simulator (Procgen). The pretrain data contains less visual complexity compared to real world data. I encourage author to explore pretraining with real world data (like MC YouTube videos [1] and first-view driving videos [2]). Or, the authors should at least make the pretraining data different from the RL observation data to simulate the difference between pretrain and finetune scenario. One way could be using different backgrounds in Procgen.
2. In my opinoin, this paper could benefit from including references to a relevant work [2] which uses IDM to label the dataset and conduct contrastive learning for policy pretraining.
3. I think this paper could benefit from additional comparison with [1] and [2]. It could provide additional insight of how methods that use extra data for IDM training perform compared to the proposed method. I am particularly interested in understanding whether training IDM  yields performance improvements or if the proposed method (without IDM) is already capable of achieving comparable, or even superior, results to the aforementioned prior works [1][2].

[1] Video PreTraining (VPT): Learning to Act by Watching Unlabeled Online Videos. https://github.com/openai/Video-Pre-Training

[2] Learning to drive by watching youtube videos: Action-conditioned contrastive policy pretraining. https://github.com/metadriverse/ACO

**Questions:**

1. Do you think this method can generalize to domains with continuous action space?
2. Does the number of discrete tokens in VQ matter? I would appreciate a ablation on this.

---

> ### Author Response · Authors · 2023-11-20
> **Response to reviewer NDab (part 1 of 2)**
>
> Dear Reviewer NDab,
> We sincerely thank you for your valuable feedback on our paper as well as your suggestions for improving it. We address your questions and comments below.
>
> > Do you think this method can generalize to domains with continuous action space?
>
>
> **We updated the paper to include results on two `dm_control` tasks — please see Figure 9 in the Appendix.** We note that the performance characteristics of LAPO in this setting are similar to our earlier Procgen results, as **LAPO significantly outperforms the PPO-from-scratch baseline.** We aim to provide results for additional continuous control environments in the camera-ready version.
>
> More generally, we note that the latent action space in LAPO is used to describe observed changes from one observation to the next and is thus actually more closely related to the task’s observation space than action space. In Procgen, where we demonstrate the success of our method, the observation space is continuous.
>
> We would also like to highlight that many prior works have successfully used discrete action representations in the context of continuous control. For example, `[1]` and `[2]` use manual, per-dimension quantization schemes for actions and `[3]` goes further by replacing Gaussian policies with Bernoulli policies, in a setting called bang-bang control. Several works learn to quantize continuous actions via vector-quantization, including `[4]` which learns a discrete action representation for use in downstream offline-RL and `[5]`, which applies vector quantization to high-level latent actions. Furthermore, `[6]` learns a quantization scheme from action-labeled demonstrations.
>
>
> In summary, we are confident that the approach taken by LAPO can be applied to many environments featuring continuous action spaces.
>
> `[1]` Janner et al. Offline Reinforcement Learning as One Big Sequence Modeling Problem. NeurIPS 2021.
> `[2]` Tang et al. Discretizing Continuous Action Space for On-Policy Optimization. AAAI 2020.
> `[3]` Seyde et al. Is Bang-Bang Control All You Need? Solving Continuous Control with Bernoulli Policies. NeurIPS 2021.
> `[4]` Luo et al. Action-Quantized Offline Reinforcement Learning for Robotic Skill Learning. CoRR 2023.
> `[5]` Jiang et al. Efficient Planning in a Compact Latent Action Space. ICLR 2023.
> `[6]` Dadashi et al. Continuous Control with Action Quantization from Demonstrations. ICML 2022.
>
>
> > Does the number of discrete tokens in VQ matter? I would appreciate a ablation on this.
>
> Yes, the number of discrete tokens that the latent policy generates does matter. When the number of tokens is large, the outcome is similar the No-VQ ablation in the paper (see Figure 3 and Figure 7). This is because the IDM is then able to pass large amounts of information to the FDM and does not need to learn to compress the transition information, thereby effectively negating the bottlenecking effect of vector quantization. When the number of tokens is too small, the performance decreases too. This is because the IDM is then no longer able to pass sufficient information to the FDM. This is particularly noticeable in environments with partial observability, such as Procgen tasks where the camera moves and shows parts of the game in future frames that weren’t visible in the past since in these environments there is a large amount of information that needs to be passed to the FDM.
>
> > Toy experimental setup: The paper motivates from learning (pretraining) with large scale web data. However, during experiments, the author uses the data from a simulator (Procgen). The pretrain data contains less visual complexity compared to real world data. I encourage author to explore pretraining with real world data (like MC YouTube videos [1] and first-view driving videos [2]). Or, the authors should at least make the pretraining data different from the RL observation data to simulate the difference between pretrain and finetune scenario. One way could be using different backgrounds in Procgen.
>
> We generally agree and intend to address the evaluation of LAPO against real-world observational data in future work. However we believe that our use of a synthetic benchmark does not reduce the utility or novelty of our work. Similar criticism could be leveled against dozens of papers that used Procgen as an evaluation benchmark, many of which were instrumental in developing the current state-of-the-art methods in RL. Indeed, Procgen was developed to be a useful evaluation setting for developing new methods in deep RL and is an effective tool to rapidly perform experiments in a tightly controlled setting. Likewise, the VPT work itself is based on a video-game environment that only loosely resembles the real world.

---

> ### Author Response · Authors · 2023-11-20
> **Response to reviewer NDab (part 2 of 2)**
>
> > In my opinoin, this paper could benefit from including references to a relevant work [2] which uses IDM to label the dataset and conduct contrastive learning for policy pretraining.
>
> Thank you for this suggestion. We agree that this is relevant prior work and included it in the related works section of the updated paper.
>
> > I think this paper could benefit from additional comparison with [1] and [2]. It could provide additional insight of how methods that use extra data for IDM training perform compared to the proposed method. I am particularly interested in understanding whether training IDM yields performance improvements or if the proposed method (without IDM) is already capable of achieving comparable, or even superior, results to the aforementioned prior works [1][2].
>
> While VPT `[1]`, ACO `[2]`, and LAPO seem to tackle a similar problem on the surface, we want to strongly emphasize that the intention behind the design of LAPO is fundamentally different. Our primary design consideration with LAPO was to push the need for true action labels as far back in the pretraining/finetuning pipeline as possible. While other methods like VPT, ACO, SS-ORL `[3]`, and RLV `[4]` need action labels to even train an IDM, not to mention a policy, LAPO can train both of these, as well as an FDM, with no need for action labels.
>
> We believe this to be a very useful feature of LAPO going forward, as it could allow us to train large-scale general-purpose latent policies on vast amounts of data from the web for which truly no action-labeled data (or even an RL environment) exists and then decode these policies to a true action space using a comparatively minute amount of labeled data from potentially only a small subset of the tasks present in the pretraining data.
>
> `[1]` Baker et al. Video PreTraining (VPT): Learning to Act by Watching Unlabeled Online Videos. NeurIPS 2022.
> `[2]` Zhang et al. Learning to Drive by Watching YouTube Videos: Action-Conditioned Contrastive Policy Pretraining. ECCV 2022.
> `[3]` Zheng et al. Semi-Supervised Offline Reinforcement Learning with Action-Free Trajectories. ICML 2023.
> `[4]` Schmeckpeper et al. Reinforcement Learning with Videos: Combining Offline Observations with Interaction. CoRL 2020.

---

> > ### Comment · Reviewer_NDab · 2023-11-21
> >
> > Thanks the authors for clarification and providing additional experimental results. My concerns are addressed. Therefore, I would like to raise my score.

---

### Official Review · Reviewer_BaGY · 2023-10-27

**Soundness:** 3 good
**Presentation:** 4 excellent
**Contribution:** 2 fair
**Rating:** 8
**Confidence:** 4

**Summary:**

In this paper, authors present a method to learn from pure observation-only data (e.g., videos demonstrating a control task without explicit action labels). The proposed algorithm LAPO aims to infer actions taking place between two consecutive observations of such data. LAPO models latent representations for the actions instead of trying to infer the true ground-truth action. It does so using two components -- inverse dynamics model (IDM) and forward dynamics model (FDM). The IDM's task is to generate a latent representation of the action that takes place between a given history of observations (including current observation) and the next observation. Consequently, the FDM's task is to generate the next observation given the current observation and a latent representation generated by IDM. Both these components are trained in conjugation to learn latent representations of unknown actions. These representations are further finetuned using true actions via either online rollout or matching actions in an offline dataset. The paper presents results on 16 discrete-action ProcGen environments.

**Strengths:**

I find the following strengths of the paper:

The paper does a superb job in terms of the writing and the clarity of the presentation. The algorithm design seems logical from the description. The UMAP plots on 16 different environments are very well-organized and exciting to go through. They convincingly substantiate the algorithm's usefulness in inferring unknown actions.

The problem statement of the work, inferring actions from observation-only data, is quite relevant for current RL research where there is a need to learn control aspects from unlabelled video demonstrations.

**Weaknesses:**

I find the following weaknesses in the paper:
1. The utility of learned latent representations for large-scale pretraining: It is evident from the plots that the actions are meaningfully clustered. However, if one has enough compute, to me, it seems plausible to train a transformer autoregressively to generate the next observations in observation-only data, ensuring that the latent representation for actions is also learned. I raise this point because if such a pretraining is possible directly with transformers, the utility of LAPO reduces. Plus, if someone wants to apply the LAPO (IDM + FDM) approach, it could be inflexible compared to a transformer extension. Anyway, I do acknowledge that LAPO does provide evidence that it is possible to meaningfully identify actions in observation-only data.

2. Issue with NOOP: From the IDM-FDM perspective, the NOOP being a null action, its latent representation should produce no transition when sent to FDM. It was unclear from the writing if there is any experimental validation of the same. Also, there is an issue with NOOP and actions with delayed effects. The present IDM-FDM model will fall short in modeling the two separately. (Authors briefly touch up on modeling the delayed effect actions, but I did not find their mention contrasting them with NOOP's.)

3. LAPO with continuous action environments: The current implementation of LAPO involves using vector quantization (VQ). However, using VQ would, in principle, limit the actions to be chosen from a finite set of discrete codes. This, in turn, creates issues while scaling to continuous action environments. The ProcGen games, environments used in the experiments, are discrete action environments, too, and I find it difficult to see how the approach would scale to non-discrete real-world control tasks.

**Questions:**

In the context of the aforementioned weaknesses, I have the following questions:

1. How can we advantageously use the LAPO latent representations for large-scale pretraining on unlabelled videos of control tasks?
2. Do authors observe that NOOP IDM representation does not affect FDM's transition?
3. Do NOOP IDM representations get clustered similarly to other actions?
4. How does NOOP compare with actions with delayed effects?
5. How will VQ-based LAPO fare in continuous action environments?
6. What are the possible extensions to the current architecture that will allow us to use LAPO seamlessly on real-world continuous action control tasks?

Given these questions, I am presently inclined to borderline reject the work. But with clarifications provided, I would definitely like to increase the score.

---

> ### Author Response · Authors · 2023-11-20
> **Response to reviewer BaGY (part 1 of 2)**
>
> Dear Reviewer BaGY,
> We sincerely thank you for your valuable feedback on our paper. We particularly appreciate your recognition of the relevance of our method to current RL research as well as the support for our experimental results as seen in the UMAP projection plots. We address your questions below.
>
> > 1. How can we advantageously use the LAPO latent representations for large-scale pretraining on unlabelled videos of control tasks?
>
> *(this answer refers both to the question 1 above as well as the related point 1 in the weaknesses section)*
>
> Our experiments used simple non-recurrent architectures to represent the IDM, FDM, and policy, since these were entirely sufficient to achieve good results in Procgen and demonstrate the usefulness of our method. However, the ideas behind LAPO are agnostic to the specific architectures used. In more complex environments it could very well be advantageous to implement these models with a large transformer. Referring to your statement "it seems plausible to train a transformer autoregressively to generate the next observations in observation-only data, **ensuring that the latent representation for actions is also learned**", we thus argue that LAPO is *exactly* a solution to this problem. Given a transformer that predicts future observations based on past observations, we would still require some additional method to extract meaningful latent actions from it. LAPO is such a method.
>
> We strongly believe that a transformer-based IDM and FDM could allow LAPO to scale to much more difficult tasks by allowing it to model complex temporal dependencies. Thus, we fully agree scaling LAPO to more powerful, transformer-based models is a promising direction for future work.
>
> > 2. Do authors observe that NOOP IDM representation does not affect FDM's transition?
>
> We agree with your point 2 in the weaknesses section, that from the perspective of the IDM & FDM, the NOOP action is simply an action like any other. However **we disagree that the NOOP action should produce no transition when sent to the FDM and argue that the NOOP action’s effect depends on the environment.** In Procgen "starpilot" for example, the NOOP action causes the agent to continue moving across the screen at the current velocity. In other Procgen tasks, the physics of the environment as well as the enemy characters similarly continue to affect the state of the environment, even under the NOOP action. Empirically, we observe that the NOOP action is modeled as expected and generally does not affect the transition in tasks such as "maze" and "miner", while indeed affecting the transition in the expected way in tasks like "starpilot". Please see the answer to the next question for evidence of the latent clustering behavior of the NOOP action.
>
> > 3. Do NOOP IDM representations get clustered similarly to other actions?
>
> Yes, similar to other actions, the NOOP action also takes on distinct clusters in the latent action space. However, it is important to note that in Procgen tasks, there are actually three kinds of actions that can act as a NOOP action:
>
> 1. The actual NOOP action
> 2. Any action that is invalid in the given environment. For example, the task "maze" only supports the actions LEFT, RIGHT, UP, and DOWN. Procgen internally maps all other actions to NOOP.
> 3. Any action that is invalid/impossible to perform in a specific state & environment. For example if the agent has a wall immediately to its right, performing the "move right" action is identical to performing the NOOP action.
>
> From the perspective of our action-free IDM & FDM, these three kinds of actions are indistinguishable from one another and therefore end up in the same latent action cluster. Our reason for omitting the NOOP action clusters in Figures 5. and 7. was simply visual clarity, since the NOOP clusters are not color-coded with a single true action label, but rather with all action labels corresponding to the cases 1-3 above.
>
> Note that this clustering behavior also highlights the strength of LAPO’s learned latent action representations, which tends to cluster actions that are more closely related in actual effect on the environment, without relying on the externally-defined action space, which might contain many redundant or equivalent actions.
>
> **Please see Figure 8 in the Appendix of the updated paper for UMAP projection plots that also include the NOOP action.**

---

> ### Author Response · Authors · 2023-11-20
> **Response to reviewer BaGY (part 2 of 2)**
>
> > 4. How does NOOP compare with actions with delayed effects?
>
> The latent action space in LAPO models the *observable effects* of the agent’s actions, rather than the actions themselves. In this way, latent actions can be thought of as highly-compressed descriptions of observed transitions.
>
> In the case of an action $a$ that is taken by the agent at time $t$, but only generates a visible effect at a later time $t+k$, the IDM will be unaware of the action being taken at time $t$ and will only observe the delayed effect. Since at time $t$, true actions $a$ and NOOP are indistinguishable from the IDM’s point of view, the latent action at that time will be the same. As the latent policy is trained to imitate the IDM’s predictions, the policy will generate the NOOP latent action at time $t$, and a latent action describing the visible effect at time $t+1$. We don’t believe this to be a significant issue, since in most environments, actions that have any impact on the state of the environment will usually elicit some degree of immediate change in the observation.
>
> > 5. How will VQ-based LAPO fare in continuous action environments?
>
> **We updated the paper to include results on two `dm_control` tasks—please see Figure 9 in the Appendix.** We note that the performance characteristics of LAPO in this setting are similar to our earlier Procgen results, as **LAPO significantly outperforms the PPO-from-scratch baseline**. We aim to provide results for additional continuous control environments in the camera-ready version.
>
> More generally, we note that the latent action space in LAPO is used to describe observed changes from one observation to the next and is thus actually more closely related to the task’s observation space than action space. In Procgen, where we demonstrate the success of our method, the observation space is continuous.
>
> We would also like to highlight that many prior works have successfully used discrete action representations in the context of continuous control. For example, `[1]` and `[2]` use manual, per-dimension quantization schemes for actions and `[3]` goes further by replacing Gaussian policies with Bernoulli policies, in a setting called bang-bang control. Several works learn to quantize continuous actions via vector-quantization, including `[4]` which learns a discrete action representation for use in downstream offline-RL and `[5]`, which applies vector quantization to high-level latent actions. Furthermore, `[6]` learns a quantization scheme from action-labeled demonstrations.
>
> In summary, we are confident that the approach taken by LAPO can be applied to many environments featuring continuous action spaces.
>
> `[1]` Janner et al. Offline Reinforcement Learning as One Big Sequence Modeling Problem. NeurIPS 2021.
> `[2]` Tang et al. Discretizing Continuous Action Space for On-Policy Optimization. AAAI 2020.
> `[3]` Seyde et al. Is Bang-Bang Control All You Need? Solving Continuous Control with Bernoulli Policies. NeurIPS 2021.
> `[4]` Luo et al. Action-Quantized Offline Reinforcement Learning for Robotic Skill Learning. CoRR 2023.
> `[5]` Jiang et al. Efficient Planning in a Compact Latent Action Space. ICLR 2023.
> `[6]` Dadashi et al. Continuous Control with Action Quantization from Demonstrations. ICML 2022.
>
> > 6. What are the possible extensions to the current architecture that will allow us to use LAPO seamlessly on real-world continuous action control tasks?
>
> We believe that the current architecture of LAPO is already well suited for continuous control tasks, as demonstrated in our `dm_control` results in Figure 9 of the updated paper. To apply LAPO seamlessly in real-world settings we are interested in the following two research directions.
>
> 1. Compared to many continuous control tasks, Procgen requires less temporal processing and can be solved close to optimally with policy architectures that simply stack the 4 most recent observations ("frame-stacking"). To apply LAPO in challenging control tasks, the use of more powerful transformer-based architectures for IDMs, FDMs and policies could be fruitful. This would also address the challenge of partial observability, as recurrence would allow the FDM to attend to the entire history of observations.
> 2. Real-world settings might feature diverse data distributions including cases where the pre-training data and fine-tuning data consists of different sets of tasks. Future work should address this multi-task transfer setting in order to be able to leverage unlabeled data from the broadest set of tasks during pre-training and then fine-tune/decode the latent policy with the much smaller set of tasks for which action-labeled data or an environment is available.

---

> > ### Comment · Reviewer_BaGY · 2023-11-21
> >
> > Thanks a lot for the detailed answers to my questions. They satisfactorily clear my doubts regarding LAPO. I am increasing my rating.

---

### Official Review · Reviewer_Z13u · 2023-11-01

**Soundness:** 3 good
**Presentation:** 3 good
**Contribution:** 3 good
**Rating:** 8
**Confidence:** 3

**Summary:**

In this paper, the authors focus on the problem of inferring and leveraging latent actions to train RL models from unlabeled demonstrations. To this end, they propose Latent Action Policies from Observation, where an Inverse Dynamics Model infers an aggressively bottlenecked latent from two observations adjacent in time. These latent actions can then be used to train a behavioral cloning model. They evaluate their approach on the Procgen dataset, show large performance margins over baselines, and analyse the learned action space visually through UMAP projection.

**Strengths:**

1. Unsupervised learning has been successfully applied to domains such as language understanding and computer vision, but this is still a frontier problem in the reinforcment learning community. This paper is a promising step in that direction.

2. The approach is simple and makes intuitive sense. A latent action could well be considered the "difference" between two observations.

3. Experimental results are strong, and the analysis of learned latent space is insightful.

**Weaknesses:**

1. The approach is only evaluated in one environment, Procgen. It is unclear if this approach will generalize across different domains (i.e. continuous control/Robotics). Additional experiments in other environments could strengthen the case of the paper.

**Questions:**

1. Do the results of LAPO generalize to other environments, especially in continuous control settings? (such as MoJoCo, DMControl, Meta-World, etc.)

---

> ### Author Response · Authors · 2023-11-20
> **Response to reviewer Z13u**
>
> Dear Reviewer Z13u,
> We sincerely thank you for your valuable feedback on our paper. We particularly appreciate your support of the strength of our experiments. We address your question below.
>
> > Do the results of LAPO generalize to other environments, especially in continuous control settings? (such as MoJoCo, DMControl, Meta-World, etc.)
>
> **We updated the paper to include results on two `dm_control` tasks—please see Figure 9 in the Appendix.** We note that the performance characteristics of LAPO in this setting are similar to our earlier Procgen results, as LAPO significantly outperforms the PPO-from-scratch baseline. We aim to provide results for additional continuous control environments in the camera-ready version.
>
> More generally, we note that the latent action space in LAPO is used to describe observed changes from one observation to the next and is thus actually more closely related to the task’s observation space than action space. In Procgen, where we demonstrate the success of our method, the observation space is continuous.
>
> We would also like to highlight that many prior works have successfully used discrete action representations in the context of continuous control. For example, `[1]` and `[2]` use manual, per-dimension quantization schemes for actions and `[3]` goes further by replacing Gaussian policies with Bernoulli policies, in a setting called bang-bang control. Several works learn to quantize continuous actions via vector-quantization, including `[4]` which learns a discrete action representation for use in downstream offline-RL and `[5]`, which applies vector quantization to high-level latent actions. Furthermore, `[6]` learns a quantization scheme from action-labeled demonstrations.
>
> In summary, we are confident that the approach taken by LAPO can be applied to many environments featuring continuous action spaces.
>
> `[1]` Janner et al. Offline Reinforcement Learning as One Big Sequence Modeling Problem. NeurIPS 2021.
> `[2]` Tang et al. Discretizing Continuous Action Space for On-Policy Optimization. AAAI 2020.
> `[3]` Seyde et al. Is Bang-Bang Control All You Need? Solving Continuous Control with Bernoulli Policies. NeurIPS 2021.
> `[4]` Luo et al. Action-Quantized Offline Reinforcement Learning for Robotic Skill Learning. CoRR 2023.
> `[5]` Jiang et al. Efficient Planning in a Compact Latent Action Space. ICLR 2023.
> `[6]` Dadashi et al. Continuous Control with Action Quantization from Demonstrations. ICML 2022.

---

> > ### Comment · Reviewer_Z13u · 2023-11-22
> >
> > Thank you for these additional experiments. They strengthen the case for the paper.

---

### Official Review · Reviewer_ATYE · 2023-11-02

**Soundness:** 3 good
**Presentation:** 3 good
**Contribution:** 3 good
**Rating:** 8
**Confidence:** 4

**Summary:**

The paper proposes an approach (LAPO) to learn latent-action policies from action-free demonstrations. The approach involves training a inverse dynamics model (IDM) along with a Forward Dynamics Model (FDM) using an unsupervised objective. The IDM learns to predict the latent action given the past and future observation. FDM on the other hand uses the predicted latent action, and past action to predict the future observation. Training these two models together for predictive consistency helps learn latent actions that can reliably explain observed transitions. To prevent the model from compressing all of past observation’s information into the latent action, the authors propose using vector quantization which acts as an information bottleneck. Finally, the authors show that latent actions can be mapped to real actions using online RL policy fine-tuning, or by using an offline dataset of action-labeled transitions. The authors show results on the Procgen benchmark. They show that the approach exceeds expert demonstrations by fine-tuning a policy pretrained with LAPO using significantly fewer number of steps. The authors also show through many visualisations that the structure of the latent space is highly interpretable

**Strengths:**

1. Learning policies from unlabeled (lacking action annotations) demonstrations is an important problem because it allows to leverage large collection of videos available on the web. The proposed approach shows how we can leverage this data to train policies in an unsupervised way, even when no labeled demonstrations are available.
2. I really appreciate the UMAP projection visualisations in the paper! The visualisations clearly demonstrate the effectiveness of the approach to learn disentangled latent space for actions, that can be meaningfully mapped to real actions corresponding to the transitions. Comparing Figure 5 and Figure 7, also clearly demonstrate the usefulness of Vector Quantization in their method.
3. The paper is well-written! I specially appreciate a well-written related works section that did a great job putting the paper in context with other related works!

**Weaknesses:**

### Weaknesses

1. The experiments assume that the underlying action space in the videos used for pre-training is the same as the action space of the agent. In other words, the videos used during pre-training are of the same agent. I think this assumption is limiting. To leverage large-scale video data available on the internet, one of the key ingredients is learning from videos that doesn’t necessarily match the action space of the agent. Imagine learning from Ego4D like ego-centric observations, and using it to execute tabletop rearrangement tasks. Or using RealEstate10K videos to learn how to navigate in indoor environments.
2. Secondly, it would have also been nice to consider experiments on continuous control tasks. The approach uses a vector quantised latent space to model actions. Will such an approach also work for actions that are continuous control?
3. Finally, the paper doesn’t fully justify using vector quanitization approach to learn disentagled latent actions. While the empirical results do show the efficacy of this approach, I don’t fully grasp why vector quantization based information bottleneck works. Additionally, did the author consider an approach like maximizing the conditional mutual information I(o_t+1 | a_t) while minimizing  I(o_t+1 | o_t)  (as in  Deep Variational Information Bottleneck, Alemi et al, 2016) to learn a latent action space that is consistent with the transition observed without over-relying on the past observations

**Questions:**

Please see weaknesses section.

---

> ### Author Response · Authors · 2023-11-20
> **Response to reviewer ATYE (part 1 of 2)**
>
> Dear Reviewer ATYE,
> We sincerely thank you for your valuable feedback on our paper. We particularly appreciate the recognition of our method’s ability to learn disentangled and interpretable latent action spaces, as demonstrated in our UMAP projections. We address your questions and comments below.
>
> > The experiments assume that the underlying action space in the videos used for pre-training is the same as the action space of the agent. In other words, the videos used during pre-training are of the same agent. I think this assumption is limiting. To leverage large-scale video data available on the internet, one of the key ingredients is learning from videos that doesn’t necessarily match the action space of the agent. Imagine learning from Ego4D like ego-centric observations, and using it to execute tabletop rearrangement tasks. Or using RealEstate10K videos to learn how to navigate in indoor environments.
>
>
> We agree that being able to leverage data from a variety of domains and tasks is crucially important for learning from large-scale video data. While we believe this to be out of scope for this current paper, we are confident that our method could be well applicable to this multi-task transfer setting. **During the IDM/FDM and latent behavior cloning training stages we make no strong assumptions on the true action space of the data-generating policy.** Even if the action space (and possibly observation space) of the downstream RL task is different from the environment of the data-generating policy, the learned latent policy might still capture information useful for tackling this different environment. We believe this multi-task setting to be a promising line of research for future work extending LAPO.
>
> > Secondly, it would have also been nice to consider experiments on continuous control tasks. The approach uses a vector quantised latent space to model actions. Will such an approach also work for actions that are continuous control?
>
> Thank you for this suggestion. **We updated the paper to include results on two `dm_control` tasks—please see Figure 9 in the Appendix.** We note that the performance characteristics of LAPO in this setting are similar to our earlier Procgen results, as LAPO significantly outperforms the PPO-from-scratch baseline. We aim to provide results for additional continuous control environments in the camera-ready version.
>
> More generally, we note that the latent action space in LAPO is used to describe observed changes from one observation to the next and is thus actually more closely related to the task’s observation space than action space. In Procgen, where we demonstrate the success of our method, the observation space is continuous.
>
> We would also like to highlight that many prior works have successfully used discrete action representations in the context of continuous control. For example, `[1]` and `[2]` use manual, per-dimension quantization schemes for actions and `[3]` goes further by replacing Gaussian policies with Bernoulli policies, in a setting called bang-bang control. Several works learn to quantize continuous actions via vector-quantization, including `[4]` which learns a discrete action representation for use in downstream offline-RL and `[5]`, which applies vector quantization to high-level latent actions. Furthermore, `[6]` learns a quantization scheme from action-labeled demonstrations.
>
>
> In summary, we are confident that the approach taken by LAPO can be applied to many environments featuring continuous action spaces.
>
> `[1]` Janner et al. Offline Reinforcement Learning as One Big Sequence Modeling Problem. NeurIPS 2021.
> `[2]` Tang et al. Discretizing Continuous Action Space for On-Policy Optimization. AAAI 2020.
> `[3]` Seyde et al. Is Bang-Bang Control All You Need? Solving Continuous Control with Bernoulli Policies. NeurIPS 2021.
> `[4]` Luo et al. Action-Quantized Offline Reinforcement Learning for Robotic Skill Learning. CoRR 2023.
> `[5]` Jiang et al. Efficient Planning in a Compact Latent Action Space. ICLR 2023.
> `[6]` Dadashi et al. Continuous Control with Action Quantization from Demonstrations. ICML 2022.

---

> ### Author Response · Authors · 2023-11-20
> **Response to reviewer ATYE (part 2 of 2)**
>
> > 3. Finally, the paper doesn’t fully justify using vector quantization approach to learn disentangled latent actions. While the empirical results do show the efficacy of this approach, I don’t fully grasp why vector quantization based information bottleneck works. Additionally, did the author consider an approach like minimizing the conditional mutual information I(a_t, o_t+1 | o_t) (as in Deep Variational Information Bottleneck, Alemi et al, 2016) to learn a latent action space that is consistent with the transition observed without over-relying on the past observations
>
> Vector-quantization (VQ) reduces the amount of information that the IDM is able to pass to the FDM down to a small set of discrete tokens. This reduces overfitting, as the IDM can no longer use a different highly-specific latent action to precisely describe each transition in the dataset, but instead is forced to learn a small set of actions that capture transition information at a high level. Intuitively, this means that instead of being able to represent pixel-level details such as “pixel X changes from color A to color B”, the IDM has to encode high-level information such as “the player character moves to the right”.
>
> This hypothesis is supported by several observations reported in our paper:
>
> 1. When not using VQ (as in the No-VQ-ablation in the paper — see Figure 7 and Figure 3 right), we observed that during IDM/FDM training the next-observation reconstruction loss was significantly lower, even though the final policy evaluation performance was worse. This indicates that the IDM is able to pass more useful information to the FDM, but the learned latent action representation is less useful in the downstream policy learning task.
> 2. To measure the quality of a learned latent action representation, we used a metric intended to measure how predictive latent actions are of true actions. This metric was computed by training a small latent -> true action decoder network and evaluating its performance on a hold-out test set. In the NO-VQ ablation, this metric was significantly worse. This too indicates that even though the NO-VQ latent action was more useful to the FDM for minimizing the pixel-level reconstruction loss, it was worse at capturing high-level action information useful for inferring the true action.
> 3. Comparing Figure 5 and Figure 7 in the paper shows that VQ reduces fragmentation of the latent action space. This fragmentation likely corresponds to the IDM using different latents for analogous transitions in different parts of the state space. For example, in Procgen tasks with randomized background images, the IDM might use a different latent for describing “the player character moves to the right” for each background. VQ forces the IDM to re-use the small set of discrete latents, thus leading to a disentangled representation.
>
> Thank you for the reference to `[1]`. Our approach could indeed be interpreted as maximizing $I(z_t, o_{t+1}| o_{t-k}, \ldots, o_{t})$, the mutual information between the future observation and latent action given past observations.
>
> `[1]` Alemi et al. Deep Variational Information Bottleneck. ICLR 2017.

---

### Meta-Review · Area_Chair_bhdU · 2023-12-05

**Metareview:**

The paper proposes a method to infer latent actions which are then used to learn latent-action policies purely from action-free demonstrations. Specifically, the paper is addressing pre-training in the context of Deep RL when the data does not explicitly have action labels. The paper is well motivated, and the experiments reasonably demonstrate the approach.

**Justification For Why Not Higher Score:**

While the experiments are reasonable, they are still on a toyish scenario. Plus, as recommended by reviewer NDab, the paper could benefit from more comparative analysis with existing method.

**Justification For Why Not Lower Score:**

The acceptance scores be reviewers are on the high side. (8,8,8,6).

---

### Decision · Program_Chairs · 2024-01-16

Accept (spotlight)